# Decoupling Quantile Representations from Loss Function

## Abstract

The simultaneous quantile regression (SQR) technique has been used to estimate uncertainties for deep learning models, but its application is limited by the requirement that the solution at the median quantile ($\tau = 0.5$) must minimize the mean absolute error (MAE). In this article, we address this limitation by demonstrating a duality between quantiles and estimated probabilities in the case of simultaneous binary quantile regression (SBQR). This allows us to decouple the construction of quantile representations from the loss function, enabling us to assign an arbitrary classifier $f(\boldsymbol{x})$ at the median quantile and generate the full spectrum of SBQR quantile representations at different values of $\tau$. We validate our approach through two applications: (i) detecting out-of-distribution samples, where we show that quantile representations outperform standard probability outputs, and (ii) calibrating models, where we demonstrate the robustness of quantile representations to distortions. We conclude with a discussion of several hypotheses arising from these findings.

## 1    Introduction

Deep learning models have become ubiquitous across diverse domains, and are increasingly being used for several critical applications. Common questions which arise in practice are - (a) Can this model be used on the given data input? and (b) If so, how much can one trust the probability prediction obtained? The former refers to the problem of Out-of-Distribution (OOD) detection [13, 9] and the latter refers to the problem of Calibration [10, 19, 22]. Understanding the applicability of a given deep learning model is a topic of current research [30, 6, 25, 14]. In this article we consider the quantile regression approach to answer these questions.

Quantile regression techniques [16, 17] provide much richer information about the model, allowing for more comprehensive analysis and understanding relationship between different variables. In [32] the authors show how simultaneous quantile regression (SQR) techniques can be used to estimate the uncertainties of the deep learning model in the case of regression problems. However, these techniques aren't widely used in modern deep learning based systems since [5] - (a) The loss function is restricted to be mean absolute error (MAE) or the pinball loss. This might not compatible with domain specific losses. (b) Moreover, it is difficult to optimize the loss function in presence of non-linearity. (c) Adapting the quantile regression approach for classification is also challenging due to piece-wise constant behavior of the loss function, due to discrete labels.

**Decoupling loss function and computing quantile representations:**    Consider the problem setting where a pre-trained classifier $f_\theta(\boldsymbol{x})$ (including the dataset on which it is trained) is given and we wish compute the quantile representations for detailed analysis of the pre-trained classifier. Classical approach is to retrain the classifier using the quantile loss (equation 2). However, one runs the risk

of losing important properties while retraining since pinball loss would have different properties compared to loss used for pre-training. Moreover, it is not clear how penalties (used for pre-training) given to attributes such as gender-bias etc. should change with the quantile[1]. Further, aspects like calibration error of the pre-trained network cannot be established by retraining with a different loss. Thus, one requires a more elegant solution than retraining using the pinball loss.

**Main Outcomes:** In this article we propose an approach to *decouple the construction of quantile representations from the loss function*. We achieve this by identifying the *Duality* property between quantiles and probabilities. We leverage the duality to construct the quantile-representations for any pre-trained classifier $f_\theta(\boldsymbol{x})$ in section 3.1. Such quantile representations are shown to capture the training distributions in section 4.2. We show that these representations outperform the baseline for OOD detection in section 4.4. And further verify that quantile representations can potentially identify OOD samples perfectly. We also show that probabilities arising from quantile-representations are *invariant* to distortions in section 4.3. Moreover, we see that standard post-hoc calibration techniques such as Platt-scaling fail to preserve invariance to distortions. Proof-of-concept experiments to improve OOD detection and identifying distribution shifts within the data are discussed in the appendix (supplementary material).

**Illustrating the Construction of Quantile Representations:** Before diving into the details, we illustrate the construction using a simple toy example and considering the problem of OOD detection. Figure 1a shows a simple toy example with 3 classes - $0, 1, 2$. Class 0 is taken to be out-of-distribution (OOD), while classes $1, 2$ are taken to in-distribution (ID). To get the quantile representation - (step 1) we first construct a simple classifier to differentiate classes $1, 2$, (step 2) To get a classifier at quantile $\tau$, construct $y_i^+ = I[p_i > \tau]^2$, where $p_i$ denotes the predicted probability in (step 1). Construct a classifier using the new labels $y_i^+$. Figure 1b illustrates the classifiers obtained at different $\tau$. In (step 3) concatenate the outputs (predictions) of all the classifiers at different $\tau$ to get the quantile-representations. Figures 1c and 1d demonstrate the advantage of having several classifiers as opposed to one. By aggregating (detailed later) the outputs from different classifiers, we are able to identify OOD vs ID samples (using One-Class-SVM [31]).

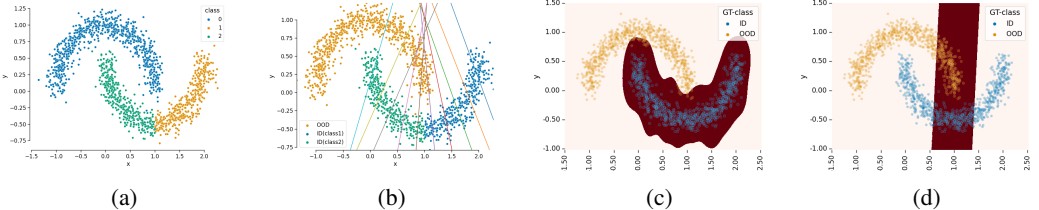

| (a) | (b) | (c) | (d) |

Figure 1: Illustrating the construction of Quantile Representations. (a) Simple toy example. (b) Illustrates different classifiers obtained for different $\tau$. (c) OOD detection using Quantile Representations. (d) OOD detection using the predictions from a single classifier. The region of In-distribution is highlighted in red. Observe that, in this case, quantile-representations are able to differentiate the in-distribution (ID) vs out-of-distribution (OOD).

**Remark:** Note that the construction in (step 2) does not depend on the procedure followed in (step 1), but only the output probabilities $p_i$. Thus, one can use *any* procedure in (step 1) without affecting (step 2). This property of quantiles is based on the duality between quantiles and probabilities ( section 2). Intuitively, quantile representations capture the distribution of the training data. Thus, given a pre-trained classifier, quantile representations can be used to analyze the classifier. In particular, as we shall shortly illustrate, one can perform calibration and OOD-detection.

## 2 Simultaneous Binary Quantile Regression (SBQR)

In this section, we review some of the theoretical foundations required for constructing quantile representations. For more details please refer to [16, 17, 32].

---

[1]Each quantile can be thought of as a significance level

[2]$I[.]$ indicates the indicator function

Let $p_{\text{data}}(X, Y)$, denote the distribution from which the data is generated. $X$ denotes the features and $Y$ denotes the targets (class labels). A classification algorithm predicts the latent variable (a.k.a *logits*) $Z$ which are used to make predictions on $Y$.

Let $\boldsymbol{x} \in \mathbb{R}^d$ denote the $d$ dimensional features and $y \in \{0, 1, \cdots, k\}$ denote the class labels (targets). We assume that the training set consists of $N$ i.i.d samples $\mathcal{D} = \{(\boldsymbol{x}_i, y_i)\}$. Let $\boldsymbol{z}_i = f_{\ell,\theta}(\boldsymbol{x}; \theta)$ denote the classification model which predicts the logits $\boldsymbol{z}_i$. In binary case ($k = 1$), applying the $\sigma$ (Sigmoid) function we obtain the probabilities, $p_i = f_\theta(\boldsymbol{x}_i) = \sigma(f_{\ell,\theta}(\boldsymbol{x}_i))$. For multi-class classification we use the $\text{softmax}(f_{\ell,\theta}(\boldsymbol{x}_i))$ to obtain the probabilities. The final class predictions are obtained using the $\arg\max_k p_{i,k}$, where $k$ denotes the class-index.

## 2.1 Review - Quantile Regression and Binary Quantile Regression

Observe that, for binary classification, $Z$ denotes a one dimensional distribution. $F_Z(\boldsymbol{z}) = P(Z \leq \boldsymbol{z})$ denotes the cumulative distribution of a random variable $Z$. The function $F_Z^{-1}(\tau) = \inf\{\boldsymbol{z} : F_Z(\boldsymbol{z}) \geq \tau\}$ denotes the quantile distribution of the variable $Z$, where $0 < \tau < 1$. The aim of quantile regression is to predict the $\tau^{th}$ quantile of the latent variable $Z$ from the data. That is, we aim to estimate $F_Z^{-1}(\tau \mid X = \boldsymbol{x})$. Minimizing pinball-loss or check-loss [16],

$$\text{pinball loss} = \sum_{i=1}^{n} \rho(f_\theta(\boldsymbol{x}_i), y_i) \quad \text{where,} \quad \rho(\hat{y}, y; \tau) = \begin{cases} \tau(y - \hat{y}) & \text{if } (y - \hat{y}) > 0 \\ (1 - \tau)(\hat{y} - y) & \text{otherwise} \end{cases} \quad (1)$$

allows us to learn $f_\theta$ which estimates the $\tau^{th}$ quantile of $Y$. When $\tau = 0.5$, we obtain the loss to be equivalent to mean absolute error (MAE). For the multi-class case we follow the one-vs-rest procedure to learn quantiles for each class.

**Simultaneous Quantile Regression (SQR):** Observe that the loss in equation 1 is for a single $\tau$. [32] argues that - minimizing the expected loss over all $\tau \in (0, 1)$,

$$\min_\psi \mathbb{E}_{\tau \sim U[0,1]}[\rho(\psi(\boldsymbol{x}, \tau), y; \tau)] \quad (2)$$

is better than optimizing for each $\tau$ separately. Using the loss in equation 2 instead of equation 1 enforces the solution to have *monotonicity property*. If $\mathcal{Q}(\boldsymbol{x}, \tau)$ denotes the solution to equation 2, monotonicity requires

$$\mathcal{Q}(\boldsymbol{x}, \tau_i) \leq \mathcal{Q}(\boldsymbol{x}, \tau_j) \Leftrightarrow \tau_i \leq \tau_j \quad (3)$$

Observe that for a given $\boldsymbol{x}_i$, the function $\mathcal{Q}(\boldsymbol{x}_i, \tau)$ can be interpreted as a (continuous) representation of $\boldsymbol{x}_i$ as $\tau$ varies over $(0, 1)$. The function $\mathcal{Q}(\boldsymbol{x}, \tau)$ is referred to as *quantile representation*. $\mathcal{Q}(\boldsymbol{x}, \tau)$ is sometimes written as $\mathcal{Q}(\boldsymbol{x}, \tau; \theta)$, where $\theta$ indicates the parameters (such as weights in a neural neural network). For brevity, we do not include the parameters $\theta$ in this article unless explicitly required.

**Remark on Notation:** To differentiate between the latent scores (logits) and probabilities - we use $\mathcal{Q}(\boldsymbol{x}, \tau)$, $f_\theta(\boldsymbol{x})$ to denote the probabilities and $\mathcal{Q}_\ell(\boldsymbol{x}, \tau)$, $f_{\ell,\theta}(\boldsymbol{x})$ to denote the latent scores. Since we have the relation $\mathcal{Q}(\boldsymbol{x}, \tau) = \sigma(\mathcal{Q}_\ell(\boldsymbol{x}, \tau))$ and $f_\ell(\boldsymbol{x}) = \sigma(f_{\ell,\theta}(\boldsymbol{x}))$ and $\sigma(.)$ is monotonic, these quantities are related by a monotonic transformation.

**Why Quantile Regression?** Quantile regression techniques are relatively unknown in the machine learning community, but offers a wide range of advantages over the traditional single point regression. Quantiles give information about the shape of the distribution, in particular if the distribution is skewed. They are robust to outliers, can model extreme events, capture uncertainty in predictions. Quantile regression techniques have been used for pediatric medicine, survival and duration time studies, discrimination and income inequality. (See supplememtary material for a more thorough discussion.)

# 3 Quantile Representations

As discussed earlier, minimizing equation 2 is not recommended due to unaccountable side-effects. Thus, we require a procedure to construct quantile representations without resorting to minimizing equation 2. In this section we present *duality* property of the quantile representations, which allows us to do this.

---
**Algorithm 1** Generating Quantile Representations.
___

- Let $\mathcal{D} = \{(\boldsymbol{x}_i, y_i)\}$ denote the training dataset. Assume that a pre-trained binary classifier $f_\theta(\boldsymbol{x})$ is given. The aim is to generate the quantile representations with respect to $f(\boldsymbol{x})$. We refer to this $f_\theta(\boldsymbol{x})$ as base-classifier.

- Assign $\mathcal{Q}(\boldsymbol{x}, 0.5) = f_\theta(\boldsymbol{x})$, that is take the median classifier to be the given classifier.

- Define $y_{i,\tau}^+ = I[f_\theta(\boldsymbol{x}_i) > (1 - \tau)]$. We refer to this as modified labels at quantile $\tau$.

- To obtain $\mathcal{Q}(\boldsymbol{x}, \tau)$, train the classifier using the dataset $\mathcal{D}_\tau^+ = \{(\boldsymbol{x}_i, y_{i,\tau}^+)\}$. Repeating this for all $\tau$ allows us to construct the quantile representation $\mathcal{Q}(\boldsymbol{x}, \tau)$.

___

## 3.1 Generating Quantile Representations Using Duality between Quantiles and Probabilities

Observe that, for binary classification, equation 1 can be written as

$$\rho(\hat{y}, y; \tau) = \begin{cases} \tau(1 - \hat{y}) & \text{if } y = 1 \\ (1 - \tau)(\hat{y}) & \text{if } y = 0 \end{cases} \tag{4}$$

Thus the following property holds :

$$\rho(\hat{y}, y; \tau) = \rho(1 - \tau, y; 1 - \hat{y}) \tag{5}$$

We refer to the above property as *duality between quantiles and probabilities*. Let $\mathcal{Q}(\boldsymbol{x}, \tau)$ denotes a solution to equation 2. It follows from above that, for a given $\boldsymbol{x}_i$ and $\tau_0$, if we have $\mathcal{Q}(\boldsymbol{x}_i, \tau_0) = p_i$, then we should also have $\mathcal{Q}(\boldsymbol{x}_i, 1 - p_i) = 1 - \tau_0$. In words, a solution which predicts the $\tau^{th}$ quantile can be interpreted as the quantile at which the probability is $1 - \tau$. This observation leads to the algorithm 1 for generating the quantile representations.

**Why does algorithm 1 return quantile representations?** Assume for an arbitrary $\boldsymbol{x}_i$, we have $\mathcal{Q}(\boldsymbol{x}_i, 0.5) = p_i$. Standard interpretation states - at quantile $\tau = 0.5$, the probability of $\boldsymbol{x}_i$ in class 1 is $p_i$. However, thanks to duality in equation 5, this can also be interpreted as - At quantile $\tau = (1 - p_i)$, the probability of $\boldsymbol{x}_i$ in class 1 is 0.5.

Thanks to monotonocity property in equation 3, we have for all $\tau < (1 - p_i)$, probability of $\boldsymbol{x}_i$ in class 1 is $< 0.5$, and hence belongs to class 0. And for all $\tau > (1 - p_i)$, probability of $\boldsymbol{x}_i$ in class 1 is $> 0.5$, and hence belongs to class 1.

This implies that at a given quantile $\tau^*$, $\boldsymbol{x}_i$ will belong to class 1 if $\tau^* > (1 - p_i)$ or if $p_i > (1 - \tau^*)$ or if $f_\theta(\boldsymbol{x}_i) > (1 - \tau^*)$. Defining, $y_{i,\tau^*}^+ = I[f_\theta(\boldsymbol{x}_i) > (1 - \tau^*)]$, we have that the classifier at quantile $\tau^*$ fits the data $\mathcal{D}_\tau^+ = \{(\boldsymbol{x}_i, y_{i,\tau^*}^+)\}$ and thus can be used to identify $\mathcal{Q}(\boldsymbol{x}, \tau^*)$. This gives us the algorithm 1 to get the quantile representations for an arbitrary classifier $f_\theta(\boldsymbol{x})$.

**Remark (Sigmoid vs Indicator function):** In theory, we approximate $\hat{y}_i = I[\hat{z}_i > 0]$ (i.e Indicator function) with the sigmoid as $\hat{y}_i = \sigma(\hat{z}_i)$. The algorithm 1 gives a solution up to this approximation. In particular we have the following theorem

**Theorem 3.1** *Let the base classifier $f_\theta(\boldsymbol{x}) = \sigma(f_{\ell,\theta}(\boldsymbol{x}))$ be obtained using the MAE loss, i.e by minimizing*

$$\min_\theta \sum_i |y_i - f_\theta(\boldsymbol{x}_i)| \tag{6}$$

*Then the solution $\mathcal{Q}(\boldsymbol{x}, \tau)$ obtained by algorithm 1 minimizes the cost in equation 2 over the dataset $\mathcal{D}$, i.e*

$$\mathcal{Q}(\boldsymbol{x}, \tau) = \underset{\psi}{\arg\min} \, \mathbb{E}_{\tau \in U[0,1]} \left[ \frac{1}{N} \sum_{i=1}^N \rho(I[\psi(\boldsymbol{x}_i, \tau) \geq 0.5], y_i; \tau) \right] \tag{7}$$

The proof for the above theorem is discussed in the supplementary material. In simple words, the proof follows from the duality and the fact that we are only interested in the labels (for this theorem) obtained by applying the indicator function.

**Importance of duality:** Algorithm 1 and theorem 3.1 hinges on the duality property. Recall that pinball loss equation 4 penalizes the positive errors and negative errors differently. In the case of binary classification, since $f_\theta(\boldsymbol{x}) \in (0,1)$, positive errors occur for class 1 and negative errors occur for class 0. Hence, the quantile value implicitly controls the probability of class 1, giving the duality property.

Thus, using quantile value as an input allows us to control the probabilities and hence confidence of our predictions. This is what we have exploited to construct quantile representations without resorting to optimizing equation 2. This ensures that the properties of the pre-trained model are preserved while still being able to compute quantile representations.

**Remark:** The other alternate to computing quantile representations are the Bayesian approaches [15]. It is known that computing the *full predictive distribution* - $p(y|\mathcal{D},x) = \int p(y|w,x)p(w|\mathcal{D})dw$ is computationally difficult. Quantile representations approximate the inverse of the c.d.f of the predictive distribution for the binary classification.

To summarize, thanks to the duality in equation 5, one can compute the quantile representations for any arbitrary classifier. This allows for detailed analysis of the classifier and the features learned. In the following section we first discuss the implementation of algorithm 1 in practice and provide both qualitative and quantitative analysis for specific models.

# 4 Experiments and Analysis

## 4.1 Generating Quantile Representations in practice

Let $f_\theta(\boldsymbol{x})$ denote a pre-trained classifier. Given a dataset $\mathcal{D} = \{(\boldsymbol{x}_i, y_i)\}_i$, we construct a *quantile dataset* - $\{(\boldsymbol{x}_i, y_{i,\tau}^+)\}_{i,\tau}$ as described in algorithm 1 with the following modifications:

- Instead of computing $y_{i,\tau}^+ = I[f_\theta(\boldsymbol{x}) > (1 - \tau)]$, we compute the $\tau^{th}$ quantile of logits - $\{f_{\ell,\theta}(\boldsymbol{x}_i)\}_i$. Moreover, as multi-class classification problem gives class imbalance under one-vs-rest paradigm, we compute *weighted-quantiles*, where weights are inversely proportional to the size of the class. Observe that since $f_{\ell,\theta}(\boldsymbol{x})$ is a monotonic function of $f_\theta(\boldsymbol{x})$, this does not make a difference in practice. However, this allows us to circumvent precision issues caused due to the sigmoid function.

- We only consider a fixed finite number of quantiles. The $n_\tau$ quantiles are given by $\left\{ {1}/{n_\tau+1}, {2}/{n_\tau+1}, \cdots, {n_\tau}/{n_\tau+1} \right\}$.

For the sake of valid experimentation, we model $\mathcal{Q}(\boldsymbol{x}, \tau)$ using the same network as $f_\theta(\boldsymbol{x})$, except for the first layer. We concatenate the value of $\tau$ to the input, resulting in slightly more number of parameters in the first layer. For efficient optimization we start the training with the weights of the pre-trained classifier $f_\theta(\boldsymbol{x})$, except for the first layer.

**Loss function to train $\mathcal{Q}_\ell(\boldsymbol{x}, \tau)$:** Recall that $\mathcal{Q}_\ell(\boldsymbol{x}, \tau)$ indicates the latent logits. We use `BinaryCrossEntropy` loss to train $\mathcal{Q}_\ell(\boldsymbol{x}, \tau)$ where the targets are given by the modified labels $\{y_{i,\tau}^+\}$.

**Inference using $\mathcal{Q}_\ell(\boldsymbol{x}, \tau)$ :** After training, we compute the probabilities as follows

$$p_i = \int_{\tau=0}^{1} I[\mathcal{Q}_\ell(\boldsymbol{x}_i, \tau) \geq 0] d\tau \approx \frac{1}{n_\tau} \sum_i I[\mathcal{Q}_\ell(\boldsymbol{x}_i, \tau) \geq 0] \tag{8}$$

**Remark:** For multi-class classification, we follow a one-vs-rest approach. Hence the loss in this case would be sum of losses over all individual classes. The probability, in multi-class case, is taken to be $\arg\max_k p_{i,k}$. Note that the probabilities $p_{i,k}$ do not add up to 1 over all classes.

**Remark:** Since the aim is to analyze the pre-trained model, we only consider one specific architecture - Resnet34, and two datasets - CIFAR10 and SVHN to illustrate our results. Other related experiments are included in the appendix (supplementary material).

**Training Details and Compute:**Training quantile representations was done on a DGX server using 4 GPUs. It takes around 10 hours (40 GPU hours in total) to learn the quantile representations for each

model. We use stochastic gradient descent with cyclic learning rate for optimization. The base_lr is taken to be $0.02$ and max_lr is taken to be $1.0$, with exponentially decreasing learning rate using $\gamma = 0.99994$. The batch_size is taken to be $1024$ for resnet34. The number of steps for the cyclic learning is taken to be $2\left(size\_dataset/2(batch\_size) + 1\right)$. The $size\_dataset$ describes the size of the training data. Complete code has been provided with the supplementary material.

## 4.2 Quantile Representations captures the distribution of the input data

Firstly, we analyze the learned quantile representations - $\mathcal{Q}_\ell(.,.)$. Broadly, the learned function $\mathcal{Q}_\ell(.,.)$ captures the 1 dimensional caricature of the input distribution, in the direction where the label changes. We illustrate this using a simple toy example (figure 2). Consider a 1-dimensional dataset in a 2d-space. The labels are assigned by splitting the dataset at the mean of the values on x-axis. We then learn a simple 1 hidden layer neural network with $100$ hidden neurons. Using this as a base classifier, we then learn the quantile representation function $\mathcal{Q}_\ell(.,.)$ as described above.

Then, we reconstruct data in the original space as follows - Assign arbitrary labels at each $\tau$ satisfying the monotonicity property equation 3. For each set of labels, keeping the learned function $\mathcal{Q}_\ell(.,.)$ fixed, learn the input which gives these labels. This is illustrated in figure 2. Observe that the shape of the learned inputs (1-d thread like structure) resembles the shape of the input dataset. This shows that the function $\mathcal{Q}_\ell(.,.)$ captures how the sample distribution changes in the input space.

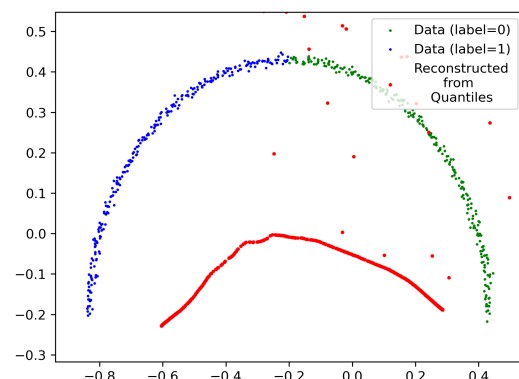

Figure 2: Quantile Representations captures the distribution of the input data distribution.

## 4.3 Calibration of ML models

For several applications the confidence of the predictions is important. This is measured by considering how well the output probabilities from the model reflect it's predictive uncertainty. This is referred to as *Calibration*.

Several methods [28, 37, 19, 1, 22] are used to improve the calibration of the deep learning models. Most of these methods consider a part of the data (apart from train data) to adjust the probability predictions. However, in [26, 23] it has been shown that most of the calibration approaches fail under distortions. In this section we show that calibration using quantile-representations are invariant to distortions.

Let $p_{i,k}$ denote the predicted probability that the sample $\boldsymbol{x}_i$ belongs to class $k$. A perfectly calibrated model (binary class) will satisfy [10] $P(\boldsymbol{y}_i = 1|p_{i,1} = p^*) = p^*$. For multi-class case this is adapted to $P(\boldsymbol{y}_i = \arg\max_k(p_{i,k})| \max_k(p_{i,k}) = p^*) = p^*$. The degree of miscalibration is usually measured using *Expected Calibration Error (ECE)*

$$E[|p^* - E[P(\boldsymbol{y} = \arg\max_k(p_{i,k})| \max_k(p_{i,k}) = p^*)]|] \tag{9}$$

This is computed by binning the predictions into $m$ bins - $B_1, B_2, \cdots, B_m$ and computing $\hat{ECE} = \sum_{i=1}^m (|B_i|/n)|\texttt{acc}(B_i) - \texttt{conf}(B_i)|$. where $\texttt{acc}(B_i) = (1/|B_i|)\sum_{j \in B_i} I[\boldsymbol{y}_j = \arg\max_k(p_{j,k})]$ denotes the accuracy of the predictions lying in $B_i$, and $\texttt{conf}(B_i) = \sum_{j \in B_i} \max_k(p_{j,k})$ indicates the average confidence of the predictions lying in $B_i$.

In the ideal scenario, we have that quantile representations predict perfectly calibrated probabilities as illustrated in the following theorem.

**Theorem 4.1** *Let $f_\theta(.)$ denote the pre-trained model. Assume that the data is generated using the model $\boldsymbol{y} = I[f_\theta(\boldsymbol{x}) + \epsilon > 0]$, where $\epsilon$ denotes the error distribution. Let $\mathcal{Q}(\boldsymbol{x}, \tau)$ denote the quantile representations obtained on this data. The probabilities obtained using equation 8 are perfectly calibrated, that is,*

$$\int_{\tau=0}^1 I[\mathcal{Q}(\boldsymbol{x}, \tau) \geq 0]d\tau = P(f_\theta(\boldsymbol{x}) + \epsilon > 0) \tag{10}$$

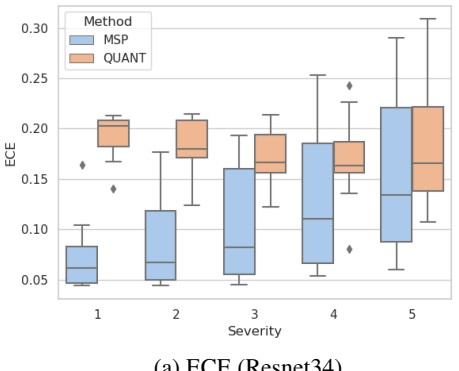
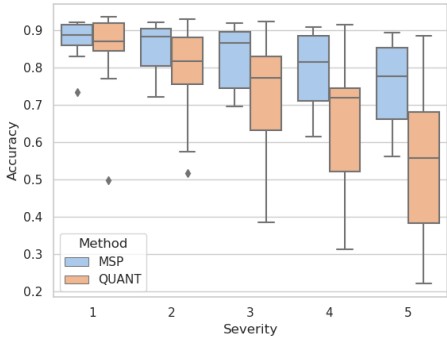

(a) ECE (Resnet34)          (b) Accuracy (Resnet34)

Figure 3: Quantile representations can be effective for calibration because they estimate probabilities using Equation equation 8, which has been shown to be robust to corruptions. As demonstrated using the CIFAR10C dataset [12], the Expected Calibration Error (ECE) of the probabilities obtained from quantile representations (QUANT) does not increase with the severity of the corruptions. In contrast, when using the standard Maximum Softmax Probability (MSP) method, the calibration error increases as the severity of the corruptions increases.

The proof for theorem 4.1 is given in the supplementary material. The main idea is the notion that $\mathcal{Q}(\boldsymbol{x}, \tau)$ captures $P(f_\theta(\boldsymbol{x}) + \epsilon < \tau)$.

Observe that, we can use theorem 4.1 to predict the calibration error of any pre-trained model, given the quantile representations. (**Remark:** This is another advantage of computing the quantile representations without retraining the original classifier. If the quantile representations are obtained by minimizing equation 2, then we cannot be sure that calibration error would remain the same.)

**Experimental Setup** In this study, we employ the CIFAR10 dataset and the Resnet34 model to investigate the robustness of classifiers. To evaluate the classifiers' robustness, we use the distorted CIFAR10 dataset introduced in [12], which contains 15 types of common corruptions at five severity levels. This dataset is a standard benchmark for testing the robustness of classifiers. We use quantile-representations trained on the CIFAR10 training data to assess the generalization performance of the classifiers on the distorted dataset. We compare the performance with Maximum Softmax Probability (MSP) as a baseline and evaluate both accuracy and calibration error. We construct the bins $\{B_i\}$ using 5 equally spaced quantiles within the predicted probabilities. The probabilities of each class are predicted using equation 8. (**Remark:** These probabilities do not add upto 1 since we consider a one-vs-rest approach.)

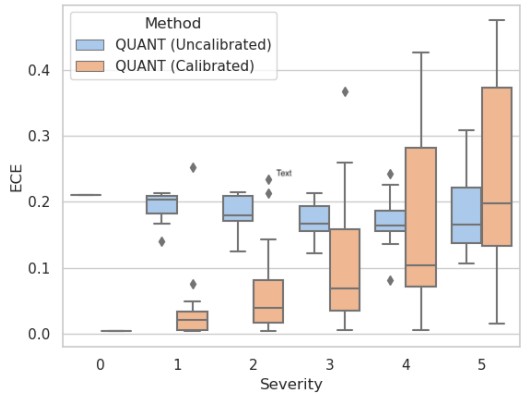

We present the results in Figure 3. As we increase the severity of the distortions, we observe that the accuracy of both the quantile representations and MSP decreases. However, the probabilities obtained from quantile representations are robust to distortions, as their Expected Calibration Error (ECE) does not increase with severity in the same way as MSP's does. This indicates that quantile representations can more accurately predict calibration error and are more resistant to distortions.

Figure 4: Correcting calibration error on the validation set may not improve performance on corrupted datasets.

**Cannot Correct the Calibration Error** Figure 3 shows that calibration error from quantile representations is robust to noise. So, an obvious question which follows is - Can we then correct it using validation data, improve the calibration

score without compromising invariance to distortions? It turns out that usual methods fail when trying to correct the calibration error of quantile representations. (**Remark:** A similar result is also obtained in Proposition 1 of [5]).

To verify this we perform the same experiment as earlier. Further we use Platt Scaling on validation data and accordingly transform the probability estimates for the corrupted datasets. These results are shown in figure 4. Observe that at severity 0, the calibration error is 0 for the corrected probabilities as expected. However, as distortion increases, the calibration error increases as well.

**Discussion:** Consider the following - Given a specific calibration error $C$ (say), let the set of all probability assignments which give the calibration error $C$ be $\mathcal{P}_C$. If the network $f_\theta(x)$ has to remain invariant to distortions, it should return one of these possible probability distributions $\mathcal{P}_C$. Our explanation for the above result is that - The vanilla neural networks do not have this property. The quantile networks, as illustrated, are evidenced to have this property. However, this also implies that calibration error *cannot* be corrected post-hoc in a sample independent manner.

## 4.4 OOD Detection using Quantile Representations

An assumption made across all machine learning models is that - Train and test datasets share the same distributions. However, test data can contain samples which are out-of-distribution (OOD) whose labels have not been seen during the training process [25]. Such samples should be ignored during inference. Hence OOD detection is a key component of reliable ML systems. Several methods [13, 20, 2] have been proposed for OOD detection. Here we check how quantile representations compare to the baseline method in [13] for OOD detection.

Quantile representations construct different classifiers at different distances from the base-classifier (illustrated in figure 1b). This allows us to differentiate between OOD samples and ID samples. Intuitively, OOD samples are far from the boundary and result in low softmax probabilities. Thus, one way to assign OOD scores to samples is by considering the maximum softmax probabilities MSP as described in [12]. We compare the OOD detection of quantile representations with this approach.

**Experimental Setup** For this study, we use the CIFAR10[18] and SVHN[24] datasets as in-distribution (ID) datasets and the iSUN[34], LSUN[36], and TinyImagenet[21] datasets as out-of-distribution (OOD) datasets. Two versions of LSUN and TinyImagenet are considered - resized to $32 \times 32$ and cropped. We evaluate the quantile representations obtained using ResNet34[11] architecture. For evaluation we use (i) AUROC: The area under the receiver operating characteristic curve of a threshold-based detector. A perfect detector corresponds to an AUROC score of 100%. (ii) TNR at 95% TPR: The probability that an OOD sample is correctly identified (classified as negative) when the true positive rate equals 95%. (iii) Detection accuracy: Measures the maximum possible classification accuracy over all possible thresholds.

**Methodology and Results** To identify OOD samples with quantile representations, we consider the entire representation - $\mathcal{Q}_\ell(x_i, \tau)$ as input features. In our experiments this would be of the dimension $n_\tau \times n\_classes$. To assign an OOD-score we use the One-Class SVM approach. The first rows of Table 1 shows the results. Observe that quantile-representations perform marginally better than than the baseline.

The more interesting result, however, is the fact that *quantile representations have all the information required to identify OOD samples.* To establish this we perform the following experiment - We use a simple linear classifier (logistic regression) to identify if the ID and OOD datasets are linearly separable or not. We measure the training accuracy to quantify linear separability - If the accuracy is close to $100\%$, then the datasets are considered to be linearly separable. For comparison we perform the same experiment with the pre-trained logits $f_{\ell,\theta}(x)$. These results are shown in the bottom rows of Table 1. Note that while the baseline scores vary with the dataset, the quantile scores are consistently close to 100%. This provides additional evidence to the hypothesis that quantile representations capture all the "relevant" information about the train distribution.

Table 1: Comparison of Quantile-Representations with baseline for OOD Detection. Observe that Quantile-Representations outperform the baseline in all the cases. The entries are represented as BASELINE/QUANTILE-REPRESENTATIONS.

| Approach | Model/ID | OOD | AUROC | TNR-TPR95 | Det.Acc |
|---|---|---|---|---|---|
| OneClassSVM | Resnet34/SVHN | iSUN | 92.28/**96.13** | 77.43/**80.15** | 89.77/**90.65** |
| | | LSUN(R) | 91.50/**95.44** | 74.95/**77.05** | 89.09/**89.67** |
| | | LSUN(C) | 92.99/**95.76** | 77.96/**80.93** | **90.10**/90.03 |
| | | Imagenet(R) | 93.52/**96.21** | **79.86**/79.60 | 90.58/**90.84** |
| | | Imagenet(C) | 94.18/**95.98** | **81.13**/79.77 | **91.23**/90.57 |
| | ResNet34/CIFAR10 | iSUN | 90.29/**93.53** | 41.90/**61.24** | 84.28/**87.06** |
| | | LSUN(R) | 90.07/**93.41** | 41.24/**61.01** | 84.25/**86.77** |
| | | LSUN(C) | 91.74/**91.79** | 45.87/**51.96** | **86.37**/85.77 |
| | | Imagenet(R) | 90.33/**92.33** | 42.18/**58.95** | 84.21/**85.47** |
| | | Imagenet(C) | 90.96/**91.44** | 43.95/**52.08** | **84.80**/84.58 |
| LinearSeparability | Resnet34/SVHN | iSUN | 83.00/**99.98** | 60.87/**99.90** | 78.98/**99.38** |
| | | LSUN(R) | 81.90/**99.98** | 56.34/**99.97** | 77.76/**99.47** |
| | | LSUN(C) | 80.44/**99.75** | 52.80/**99.25** | 75.35/**97.76** |
| | | Imagenet(R) | 80.31/**99.96** | 57.61/**99.85** | 77.19/**99.19** |
| | | Imagenet(C) | 81.88/**99.93** | 61.28/**99.78** | 78.75/**98.91** |
| | ResNet34/CIFAR10 | iSUN | 98.73/**99.94** | 96.06/**99.88** | 95.75/**98.92** |
| | | LSUN(R) | 98.80/**99.96** | 96.18/**99.84** | 95.94/**99.17** |
| | | LSUN(C) | 92.78/**99.55** | 70.72/**98.31** | 87.47/**96.87** |
| | | Imagenet(R) | 95.27/**99.74** | 86.76/**98.93** | 91.02/**97.72** |
| | | Imagenet(C) | 94.38/**99.67** | 82.37/**98.72** | 89.15/**97.25** |

## 5 Related Work

[16, 27, 29, 3] provides a comprehensive overview of approaches related to quantile regression and identifying the parameters. [4] extends the quantiles to multi-variate case. [32, 33] use quantile regression based approaches for estimating confidence of neural networks based predictions. [1, 8] uses conformal methods to calibrate probabilities, and is closely related to computing quantiles. [5] proposes a similar algorithm to overcome the restriction to pinball loss for regression problems. [7] generates predictive regions using quantile regression techniques.

## 6 Conclusion, Limitations and Future work

To summarize, in this article we show the duality between quantiles and probabilities in the case of SBQR. Exploiting the duality, we propose an algorithm to compute quantile representations for any given base classifier. We verify that the quantile representations model the training distribution well both qualitatively (by resconstructing the data in the input space) and quantitatively (using OOD detection baseline). We further show that the probabilities from quantile representations are robust to distortions. Interestingly, we found that traditional approaches cannot be used to correct the calibration error. Further experiments to validate the observations made in this article are discussed in the supplementary material.

The main limitation of the approach is the computation required for algorithm 1 for large scale datasets. Note that algorithm 1 creates $n_\tau$ copies of the same dataset by assigning different labels. For large scale datasets and large scale networks this requires a lot more computation than using a pre-trained classifier. However, we hypothesize that - we only need to retrain the quantile network only on a small sample size instead of the entire dataset We consider this for future work.

Based on strong convexity and Lipschitzness of loss functions, automatic learning rates can be computed for large networks via the inverse of the Lipschitz constant of the loss function being an ideal learning rate [35]. We conjecture that exploiting the loss functions which inherit some convexity and Lipschitz properties from the known, closed form loss representations would achieve higher learning rates for faster convergence to compute quantile representations. We defer this as future work.

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
