# OpenReview forum: "Decoupling Quantile Representations from Loss Function"
_NeurIPS.cc/2023/Conference — Submitted to NeurIPS 2023_

### Official Review · Reviewer_4AGX · 2023-06-22

**Soundness:** 2 fair
**Presentation:** 3 good
**Contribution:** 2 fair
**Rating:** 3
**Confidence:** 3

**Summary:**

This paper proposes a framework which extends an existing pre-trained classifier to a full quantile representation of the target data. This is done by setting the pre-trained classifier as the median classifier and by optimizing over pinball losses over different quantile values. Experimental results show that the proposed framework is competitive in out-of-distribution detection and calibrating models.

**Strengths:**

The paper proposes an idea which extends a pre-trained classifier to a whole quantile representation of the data. The framework is based on a simple observation which connects probability with classification quantile levels. Overall, the paper is well-written and easy to follow.


**Weaknesses:**

- The main algorithm is based on the monotonicity property. However, monotonicity is not guaranteed in general. In particular, the statement "Using eq.2 enforces the solution to have monotonicity property" is misleading. This is true only when the global minimum is obtained, which is often not the case in neural networks. Even minimizing with eq.2 can lead to a model that is not monotone, and monotonicity is enforced in model design in real practice. See, e.g. [2,4]

- In the experiments, the proposed framework is only compared with a simple baseline model. Therefore the results are not too convincing.

- I do not agree with the statement that in modern machine learning, loss functions are restricted to pinball loss or MAE. Eq.2, also called continuous ranked probability score is widely applied in quantile regression as well as machine learning.

- In order to perform algorithm 1, we need to minimize multiple pinball losses on different quantile levels (and use the data). The overall costs seem to be similar to minimizing eq.2 with the expectation estimated by sampling (similarly to eq.8) or numerical integration. Therefore I am not sure about the statement that "pinball loss/eq.2 is difficult to optimize"

- Overall, I am not too convinced that the proposed framework is more preferable than optimizing an estimated eq.2. I recommend the authors to emphasize the importance of preserving the original property of the pre-trained classifier, and to experimentally show that in some aspects, the proposed framework can compete with a quantile regressor obtained by optimizing over an estimated eq.2.

- The related work section on quantile regression tends to only include classical paper, and does not cover enough recent papers on quantile regression with machine learning. For instance, in probabilistic time series forecasting, there is a collection of work which successfully applies quantile regression with machine learning, for example, [1-3]

[1] A multi-horizon quantile recurrent forecaster by Wen et al.
[2] Probabilistic forecasting with spline quantile function RNNs by Gasthaus et al.
[3] Multi-horizon time series forecasting with temporal attention learning by Fan et al.
[4] Learning quantile functions without quantile crossing for distribution-free time series forecasting by Park et al.

**Questions:**

- The paper is based on the point that we want to preserve the original property of the pre-trained classifier. Is there an important application to this?

- Are there other similar framework which also extends a pre-trained classifier to compare with?


**Limitations:**

As mentioned in the weaknesses section, the framework is based on the monotonicity of the model. However, the monotonicity is not guaranteed in real practice.

---

> ### Author Rebuttal · Authors · 2023-08-05
>
> We thank the reviewer for the comments.
>
> **Guarantee on Monotonicity** We agree with the statement that monotonicity may not be guaranteed in practice. We only meant that eq(2) biases the solution towards having monotonicity property compared to optimising for each $\tau$ individually.
>
> In our case, observe that the way we define the labels - $I[f_{\theta}(x_i) > 1-\tau]$ is a monotonic function of $\tau$. So, the monotonicity is inbuilt into the dataset. This implicitly gives a strong penalty if monotonicity is not satisfied.
>
> We also agree that in practice, for a few data points it might happen, due to numerical or approximation issues ( since global minima might not be attained), that monotonicity is not preserved. So, we also perform a sanity check by plotting the predicted value at different $\tau$ on randomly selected samples (see figure in attached pdf, global response) to verify that monotonicity is preserved for $Q(x,\tau)$.
>
> **Evaluation**The reason for choosing the MSP baseline is - this is considered a very strong baseline for OOD detection [13]. Moreover, as reviewer zXud points out, a slight variation ignoring the softmax, called maximum logit score (MLS), gives state-of-the-art results. The additional results show that the maximum probability from quantiles are still competitive with the MLS approach as well.
>
> Please see the global response for additional results and discussion as well.
>
> **Loss function in Algorithm 1** We apologise for the confusion caused by the 4th bullet in algorithm 1. (please also see the global response )
>
> We actually use the binary cross entropy loss (BCE) (w.r.t logits) (lines 179-180) to obtain $Q(x,\tau)$, where $Q(x, \tau)$ is modelled by a DNN which takes in $(x, \tau)$ and predicts $y_{i,\tau}^{+}$. Since BCE loss is relatively easy to optimize compared to pinball loss, we claim that our approach is easier to optimize.
>
> **Why proposed method is more preferable than minimizing eq(2)** The motivation of this work is that properties of the base-classifier (pre-trained) are lost if we optimize eq(2) independently. To illustrate this statement, conider Experiment 4.3 - estimating the calibration error.
>
> To recall, the aim is - We are given a pre-trained classifier and are required to provide probabilities for this classifier ( and accordingly compute the calibration error) which are invariant with respect to distortions.
>
> Now, if we retrain this classifier using eq(2) from scratch then there is not reason to believe that the calibration error of the newly trained model would be the same as the original model. In fact, we are not using any information about the pre-trained network but are constructing the new model ignoring the pre-trained classifier.
>
> In our case, thanks to theorem 4.1, we can be sure that the calibration error computed when using probabilities from eq(8) would be the distortion-invariant probabilities of the pre-trained classifier. This is illustrated in Experiment 4.3 This would not have been possible without the framework proposed here.
>
> **Practical Application and Similar Frameworks** While we cannot explicitly state practical application of preserving the properties, we can certainly hypothesize several use cases.
>
> For instance - Consider a case where we train the classifier using a loss function which penalizes gender-bias (say) by including a penalty within the loss function. It is not clear how eq(2) should be modified to such a loss function. The naive approach of adding the penalty would not help since the penalty coefficients should be different for different $\tau$. Added to this is the complexity that $\tau$ itself is factor which penalizes positive errors vs negative errors. So, there is no clear method to retrain the classifier from scratch using eq(2). However, our approach overcomes this using the information already embedded in the pre-trained classifier.
>
> Also note that such a solution (proposed here) was not possible till date and hence could be the reason why no works exist which tackle this issue. We are also not aware of any similar frameworks which can achieve this decoupling for classification. But the work in [5] considers the problems with pinball loss for regression problems and provides a few solutions for the same.

---

> > ### Comment · Reviewer_4AGX · 2023-08-16
> > **Response of reviewer**
> >
> > I would like to thank the authors for the clarification.
> >
> > The authors have addressed why the proposed approach is more preferable than to optimize the simultaneous quantile regression loss (eq. 2). However, the fact that the monotonicity property cannot be guaranteed and inadequate comparing methods are still major weaknesses.
> >
> > After reading the rebuttal and other reviewers' comments, I will raise the score to 4: borderline reject. I suggest the authors to include more recent baselines to compare with, and rephrase the writing to indicate that the proposed method only encourage monotonicity and it does not guarantee it.

---

> > > ### Author Response · Authors · 2023-08-16
> > >
> > > We thank the reviewer for increasing the score.
> > >
> > > Please note that we have included new results (attached pdf in the global response) where we compare our approach with MLS (ICLR 2022) which is state-of-the-art. Our findings/conclusions have not changed with the inclusion of the new results as well.
> > >
> > > Also, while we agree that monotonicity cannot be guaranteed at all points, we indeed check (please see additional pdf figure 2) that this holds for some randomly selected samples. This (at least empirically) verifies the monotonicity claim.
> > >
> > > Further, techniques from both [2,4] can indeed be further incorporated into the approach and our idea is perfectly compatible with the ideas from here. However, including them would not make them comparable with the baselines we have considered. We shall include the following as a remark - "While monotonicity may not be guaranteed, it can be encoded in the architecture by considering ideas from [Gasthaus et al. , Park et al.]". We will also try to include the recent quantile regression articles within the literature review.

---

> > > > ### Comment · Reviewer_4AGX · 2023-08-19
> > > > **Reviewer response**
> > > >
> > > > I want to thank the authors for their response. I have considered what the authors mentioned in the latest response when I updated the score. While the authors have included a state-of-the-art comparing method, I believe that more state-of-the-art methods should be included for an acceptance into Neurips. I will keep my score as 4: borderline reject.

---

### Official Review · Reviewer_d48X · 2023-07-06

**Soundness:** 2 fair
**Presentation:** 2 fair
**Contribution:** 2 fair
**Rating:** 4
**Confidence:** 3

**Summary:**

This paper proposed a method to construct quantile representations of binary classification problems. This construction does not require to use the pinball loss, resolving the difficulty of optimizing the pinball loss. The paper verified its method through experiments.


**Strengths:**

The proposed approach looks novel and uses an interesting property of pinball loss.

**Weaknesses:**

The notations in this paper are confusing, making it hard for me to understand the proposed method and the proof of the theoretical result.
1. The minimizer of (1) f_\theta is claimed to be a quantile function. I am confused about which random variable this quantile function belongs to, Y or Z? Y is a binary random variable, and Z is a deterministic function of X, according to line 73-74. That means Z should not have a conditional quantile function (since it is a deterministic function of X), and the quantile function of Y can only take values in {0, 0.5, 1} (since Y is binary).
2. Eq. (2) and Eq. (7) are different. In Eq. (2), there is no indicator around \psi, whereas in Eq. (2), there is an indicator around \psi. So it is confusing whether Q should be a function inside the indicator, or a function outside of the indicator.
3. In Algorithm 1, the 4th bullet point, it is not clear which loss is used to train the classifier. Note that the loss function should be important for Theorem 3.1 to hold: Theorem 3.1 is claiming the minimizers of two empirical risk minimization problems (not population risk minimization) are equivalent, and it cannot be the case that they are equivalent no matter which loss function is used.

**Questions:**

Basically, I don’t see how this paper properly defines the conditional quantile function of a binary classification distribution. This can neither be the conditional quantile function of Z = f_{\ell, \theta}(X, \theta) given X, since f_\theta is a deterministic function; nor the conditional quantile function of Y given X, since Y is binary.

---

> ### Author Rebuttal · Authors · 2023-08-05
>
> We thank the reviewer for the comments. We have tried our best to stick to standard notation in [16]. However, since the idea is novel we had to deviate from standard notation in a few places. Hope the comments below clarify the notation.
>
> **Clarification on the problem setup:** The quantile function belongs to the variable $Z$ in lines 72-73. $Z$ is not considered as a deterministic function of $X$.
>
> Formally, the equations describing the model are $Z = f^*(X) + \epsilon$, where $f^*$ is a deterministic function (predicted approximation of ground-truth) and $\epsilon$ denotes the error distribution which introduces the randomness [16]. Hence even for the fixed $X$ the latent $Z$ might take on different values. The prediction is then obtained using $Y = I[Z \geq 0]$.
>
> **Eq(2) vs Eq(7)** It is true that eq(2) and eq(7) are different. Note that eq(2) has an implicit sigmoid function within it since the outputs are assumed to belong to $(0,1)$. In eq(7) we use an additional hard thresholding at value $0.5$, over and above the sigmoid function implicit in eq(2). This is because, theoretically we can only guarantee that the algorithm will give a minimizer upto the thresholding.
>
> In words - if you take a solution with respect to eq(2), threshold it at $0.5$, then you would get a minimizer for RHS of eq(7). The same is true for the solution from the algorithm - If you take a solution obtained using the algorithm, threshold it at $0.5$ then you would get a minimizer of RHS of eq(7). This is the statement of the theorem 3.1. Hence $Q(.,.)$ would be a function inside the indicator.
>
> Hope this clarifies the relation between eq(7) and eq(2).
>
> **Loss function in Algorithm 1** We use the usual binary cross entropy loss (w.r.t logits) (lines 179-180) to obtain $Q(x,\tau)$.  We shall include this in the algorithm as well in the revised version. Please also see the global response.
>
> While it is true that the loss function matters for the equivalence, we have a lot of flexibility in the choice of loss-function as well. Since we are only using the empirical risk and not population risk, we only need that both the functions predict the same label on the train dataset. This can be achieved by several different loss functions.

---

> > ### Comment · Reviewer_d48X · 2023-08-17
> > **Thanks for clarification**
> >
> > Thank the authors for clarification. However, I am not able to increase my score since I believe the formulation issue is fundamental and I think it requires substantial revision to make it consistent.

---

> > > ### Author Response · Authors · 2023-08-18
> > >
> > > We thank the reviewer for his response.
> > >
> > > Please note that what we described is the *latent variable model* which has been widely used, and has not been introduced by us in this article. Please see section 8.2 from [16, Koenker, Quantile Regression] on its relation to quantiles. Further, the following references use this model as well.
> > >
> > > [1] Smoothed Binary Regression Quantiles: Gregory Kordas, Journal of Applied Econometrics, Apr., 2006, Vol. 21, No. 3, pp. 387-407
> > >
> > > [2] Binary Quantile Regression: A Bayesian Approach Based on the Asymmetric Laplace Distribution: Dries F. Benoit and Dirk Van Den Poel, Journal of Applied Econometrics, November-December 2012, Vol. 27, No. 7, pp. 1174-1188
> > >
> > > We kindly request the reviewer to let us know the specific issue, so we may be able to respond to it better.

---

> > > > ### Comment · Reviewer_d48X · 2023-08-20
> > > > **Some further confusions during my reading of this paper**
> > > >
> > > > I think the equation $Z = f_*(X) + \eps$ is a key equation that hurdled my first reading of this paper. Without this equation, I am not able to understand the precise definition of the conditional quantile function (my understanding was that $z = f_{l, \theta}(x, \theta)$ is a deterministic function of $x$).
> > > >
> > > > Taking this statistical model $Z = f_*(X) + \eps$ in my mind, I re-read the paper again today. I am stuck at Theorem 3's proof, Equation (6) of Appendix A. I don't see how this equation holds from the 4th bullet point of Algorithm 1: "To obtain $Q(x, \tau)$, train the classifier using the dataset xxx". I guess the main paper 124-134 is an explanation of this equation (6), but this explanation is hard for me to follow. I feel that explaining this through equations should be helpful.
> > > >
> > > > Furthermore, I am confused about the remark in lines 135-137. It says, "In theory, we approximate xxxx with the sigmoid as xxxx. The Algorithm 1 gives a solution up to this approximation." Theorem 3.1, Eq. (7) gives an equality, instead of an approximate equality. Do the authors suggest that Eq. (7) is approximately correct?

---

> > > > > ### Author Response · Authors · 2023-08-20
> > > > >
> > > > > We thank the reviewer for this response. And we greatly appreciate any questions and discussion.
> > > > >
> > > > > **Regarding Eq (6) in Appendix A:** Note that $Q(x,\tau)$ is trained on the dataset $(x_i, y_{i,\tau}^{+})$. So, we have $Q(x_i,\tau) = y_{i,\tau\}^{+}$, assuming that we select a model with sufficient complexity. From bullet 3 of algorithm 1 we have $Q(x_i,\tau) = y_{i,\tau}^{+} = I[f_{\theta}(x_i) > 1-\tau]$. This is eq(6) in the appendix.
> > > > >
> > > > > **Lines 135-137:** The eq(7) is exact.
> > > > >
> > > > > We meant that in "general" theory, we do not directly optimise cost with indicator functions since gradient-descent does not perform well on this, and hence we use the sigmoid. But, due to arbitrariness in the scale of sigmoid function, we *cannot* guarantee that the solution obtained by algorithm 1 would be the same as the one obtained with eq(2).
> > > > >
> > > > > So, if we add an additional threshold on top of sigmoid at $0.5$ (which is eq(7)) we can guarantee that the solution obtained by algorithm 1 would be the same as the one obtained with eq(2).
> > > > >
> > > > > By stating - *solution upto this approximation* we meant that you have to add an additional thresholding on top of sigmoid, and only then we can guarantee the minimisers to be the same.
> > > > >
> > > > > Hope this clarifies the theorem.

---

> > > > > > ### Comment · Reviewer_d48X · 2023-08-20
> > > > > >
> > > > > > There seems to be a gap between $Q(x_i, \tau) = y_{i, \tau}^+ = I[f_{\theta}(x_i) > 1 - \tau]$ and Eq. (6) in the appendix which is $I[Q(x_i, \tau) >= 0.5] = I[f_\theta(x_i) > 1 - \tau]$. What did I miss here?

---

> > > > > > > ### Comment · Reviewer_d48X · 2023-08-20
> > > > > > >
> > > > > > > Oh, I see. This is trivial. Let me think more.

---

> > > > > > > > ### Comment · Reviewer_d48X · 2023-08-20
> > > > > > > >
> > > > > > > > Then what I am confused is: $Q(x_i, \tau) = y_{i, \tau}^+$ means that Q is binary. Then it does not specify a meaningful conditional quantile function, does it?

---

> > > > > > > > > ### Author Response · Authors · 2023-08-20
> > > > > > > > >
> > > > > > > > > $Q(x_i,\tau) = y_{i,\tau}^{+}$ is only true on the train dataset. This is sufficient to prove the theorem. For general $x_i$ it would take on different values.

---

### Official Review · Reviewer_EyEA · 2023-07-08

**Soundness:** 3 good
**Presentation:** 2 fair
**Contribution:** 3 good
**Rating:** 6
**Confidence:** 1

**Summary:**

This paper proposed an effective approach to estimate the uncertainty for the classification model. By decoupling the quantile representation from the loss function, the proposed method works for arbitrary base classifier.

**Strengths:**

1. this paper discovered the duality between the probabilities and quantile, and proposed a method to decouple the quantile representations from the loss function for pre-trained classification models;
1. the proposed method demonstrated effectiveness on two problems: out-of-distribution (OOD) discovery and model calibration.

**Weaknesses:**

1. major weakness, as mentioned by the authors, is the scalability of the proposed method wrt large scale datasets and models.

**Questions:**

N/A.

**Limitations:**

no potential negative societal impact of the work is noted to the best knowledge of the reviewer

---

> ### Author Rebuttal · Authors · 2023-08-05
>
> We thank the reviewer for the comments.

---

### Official Review · Reviewer_dSRG · 2023-07-09

**Soundness:** 2 fair
**Presentation:** 2 fair
**Contribution:** 2 fair
**Rating:** 5
**Confidence:** 3

**Summary:**

This paper aims to address a fundamental question: given a pre-trained classifier $f_{\theta}(x)$ and its corresponding training dataset, how can we compute quantile representations to conduct a detailed analysis of the pre-trained classifier? The authors investigate alternative approaches to this problem beyond the classical method of retraining the model using the pinball loss.

In their work, the authors adopt a straightforward approach that involves decoupling the loss function from the computation of quantile representations. By taking this approach, they propose a novel solution to efficiently analyze the pre-trained classifier without losing its important properties.

**Strengths:**


The primary contribution of the paper is centered around Section 3.1. Within this section, the authors focus on the binary classification problem and highlight the duality that exists between quantiles and probabilities in this specific scenario. They recognize this duality and propose an algorithm that harnesses the power of a pre-trained binary classifier to derive the desired quantile representation

**Weaknesses:**

The contribution of this paper is limited for the following reasons:

1. **Specific to binary classification**: The proposed algorithm for generating quantile representations is based on the duality between quantiles and probabilities, which is evident in the context of binary classification. However, this duality may not hold true for general classification problems. As a result, the applicability of the proposed algorithm is restricted to binary classification scenarios, limiting its broader impact.

2. **Unavoidable quantile crossing**: Since the optimization of quantiles for different $\tau$ values is performed independently, it is possible to observe quantile crossing in the final solution. This can introduce challenges in interpreting the results and may affect the overall performance of the proposed algorithm. Additionally, there seems to be a discrepancy between Equation (7) and the proposed algorithm. Equation (7) suggests obtaining all quantiles simultaneously, while the algorithm indicates that the quantiles are obtained separately. This discrepancy raises confusion and needs further clarification.

3. **Extensive experiments need** :Another limitation of the paper is the lack of extensive experiments. In Section 4.3, the authors mention several methods mentioned in the references that are used to enhance the calibration of deep learning models. However, these methods are not included as baselines in the subsequent experiments. As a result, the proposed method is only compared with simple MSP, which may not provide sufficient evidence to support its effectiveness and superiority.

**Questions:**

 About the toy example

The purpose of the toy example in the paper is to illustrate the construction of quantile representations, as depicted in Figure 1(b). However, the specific details regarding the generation of results in Figures 1(c) and 1(d) require further clarification. Since the classifier is only trained for class 1 and class 2, it is important to understand how the binary classifier is utilized to differentiate between inliers and outliers.

To gain a clearer understanding, more information is needed about the methodology employed in the paper. Specifically, it would be helpful to know the selected percentage $\tau$ that is used in this context. This information will provide insights into how the classifier is used to distinguish between inliers and outliers based on the quantile representations.

**Limitations:**

.

---

> ### Author Rebuttal · Authors · 2023-08-05
>
> We thank the reviewer for his comments.
>
> **Restricted to binary classification?** We do not consider this as a severe restriction since we have several ways to extend binary to multi-class classification from the literature of SVMs. We use one-vs-rest approach in this article to overcome this problem in our experiments.
>
> However, there might be more elegant approaches such as using high-dimensional quantiles instead. This shall be considered for future work.
>
> **Unavoidable quantile crossing** Sorry for the confusion caused by the wording of bullet 4 in algorithm 1. We hope to rephrase the line better as follows in the revised version:
>
> *To get the classifier $Q(x,\tau)$, collect the dataset $D_{\tau}^{+} = \{(x_i, y_{i,\tau}^{+})}$ for all $\tau$, and train a network with the new dataset. NOTE: For the input $(x_i,\tau)$ and the expected label predicted is supposed to be $y_{i, \tau}^{+}$.*
>
> Hence, we indeed simultaneously train for all $\tau$. Moreover, observe that the way labels are constructed using $I[f_{\theta}(x_i) > 1-\tau]$ is a monotonic function of $\tau$. Hence  monotonicty with respect to quantiles is in-built into the algorithm. This avoids quantile crossing. We provide a sanity check of the probabilities (in added pdf global response) to verify this as well.
>
> **Evaluation** The reason for choosing the MSP baseline is - this is considered a very strong baseline for OOD detection [13]. Moreover, as reviewer zXud points out, a slight variation ignoring the softmax results, called Maximum Logic Score (MLS), results in state-of-the-art results. We compare the maximum probabilities obtained from our approach with MLS and show that our approach is still competitive.
>
> Please see attached pdf and the global response for additional results and discussion.
>
> **Figure 1(c) and (d)** Please see the global response for clarification on how figure 1 was obtained. And how the representations are constructed. These representations are used as an input to OneClassSVM to identify the training region. And as shown in figure 1(c) these are pretty accurate.
>
> Figure1(d) uses the simple logic score of the base classifier to identify the region. That is, we use a single logit score of the base classifier as an input to OneClassSVM to identify the training region. And clearly this fails to identify the training region. Intuitively, single classifiers cannot differentiate between samples with high probability and samples which are out-of-distribution.
>
> **Selection of $\tau$(for figure 1)** To generate the boundaries we use 10 different values of $\tau$. We include a new figure where each boundary corresponding to $\tau$ is labelled in the pdf of global response.
>
> **Selection of $\tau$(in general)** We select 100 different values of $\tau$ equally spaced between 0 and 1. Please see lines 172-173 for further details.

---

> > ### Comment · Reviewer_dSRG · 2023-08-20
> >
> > I appreciate the thoughtful feedback from the authors. It's reassuring that many of my concerns have been adequately addressed, and the additional information provided has enhanced my understanding of their motivation. I observed that fellow reviewers also raised the point that monotonicity is merely encouraged, not assured. The supplementary experimental results seem to corroborate the satisfactory incorporation of monotonicity.
> >
> > In overall assessment, I find this paper to be teetering on the borderline. I'm inclined to maintain my original score.

---

> > > ### Author Response · Authors · 2023-08-20
> > >
> > > We thank the reviewer for the response.

---

### Official Review · Reviewer_zXud · 2023-07-26

**Soundness:** 2 fair
**Presentation:** 2 fair
**Contribution:** 3 good
**Rating:** 3
**Confidence:** 3

**Summary:**

This paper tackles the problem adapting quantile regression (QR) to  deep neural networks for classification tasks. QR takes into account the distribution of the dependent variable and fits a model that gives low error for all quantiles. Logistic regression is a special case of QR where only the tau=0.5 (median) quantile is considered. In previous work, application of QR is usually coupled to the loss function or the model used. In this paper, they decouple these things, so they can theoretically apply QR based learning to any loss function and any network. They establish a duality between quantiles and estimated probabilities in binary classification. Based on this duality, they train separate classifiers for different taus based on the predictions of the median (tau=0.5) classifier. Authors experimentally validate their method in two tasks: out of distribution detection and showing robustness to distortions.

**Strengths:**

+ Tackles an important problem: although quantile regression (QR) has a solid history in econometrics and statistics, it is relatively new in the machine learning community. Its application to modern deep neural networks is potentially useful and interesting.

**Weaknesses:**

- Limited evaluation. For out of distribution (OOD) detection, they compare their results only with OneClassSVM, a method from 2017. There are stronger and more recent methods in the literature. For example, in the ICLR2022 paper titled "Open-Set Recognition: A Good Closed-Set Classifier is All You Need", they compare many SOTA methods for this task and arrive to the conclusion that "maximum logit score (MLS) thresholding" is a strong method. The authors should have at least compared their results with MLS. Of course, there are many more relevant methods, which are all cited in that ICLR2022 paper.

- Language and presentation issues prevent an easy reading of this paper. A non-exhaustive list follows.
- L26: "However, these techniques aren’t widely used in modern deep learning based systems since [5]" -> what  does this mean? Is "since" the wrong word?
- L28: "might not compatible with"
- L55: "taken to in-distribution"
- Fig 1 is not easily understandable. E.g., an intuitive explanation as to why we get those classifier in 1 (b) for different taus? which classifier correspond to which tau? Transition from 1(b) to 1(c) not clear.
- L76: what is l in the subscript of f?
- In the last step of Algorithm 1, you wrote "train the classifier" but I think you mean "train a separate classifier", right?
- I found Fig2 to be completely not understandable. I cannot even point to any questions.

**Questions:**

L60: "Figures 1c and 1d demonstrate the advantage of having several classifiers as opposed to one." -> Can't other models (e.g. deep ensembles and DUQ, to cite on top my of hat) achieve the same?

Regarding Algorithm 1, what is the point of training a new classifier for a tau value? Can't you obtain it directly from $I[f_\theta (x_i)>1-\tau]$?



**Limitations:**

While listing the areas where QR is applied it would be useful to provide citations (L108).

---

> ### Author Rebuttal · Authors · 2023-08-04
>
> **Evaluation:** Please see the global rebuttal for the explanation of the choices we have made while evaluating the approach.
>
> **Language and Presentation** Sorry for the confusion caused. (i) We meant that [5] actually discusses the same shortcomings as pointed out here, (ii) We shall recheck for any grammatical errors and correct them in the revision. (iii) we use the subscript $\ell$ to differentiate logits and probability values.
>
> **Figure 1 Explanation** The aim of the figure is to illustrate what happens at different $\tau$. (1) We first identify a base-classifier by simply using logistic regression between classes $1$ and $2$. (2) Then (similar to algorithm 1) we construct a new dataset at a given $\tau$ by thresholding the probability values. (3) Then for each $\tau$ we use the new dataset to build a classifier. These are the different lines you see in figure 1b. We have updated the figure 1b to include the values of $\tau$ for each classifier in the included pdf.
>
> **From 1(b) to 1(c)** Let $x$ be any input, and say we get $y_{\tau}$ to be the corresponding logit for the classifier at $\tau$. Now, assign the representation - $[y_{0.1}, y_{0.2}, \cdots, y_{0.9}]$ to the input $x$. Use these representations (for each point in class 1 and 2) as input to OneClassSVM and identify the inliners vs outliers. The red region is marked shows inliners.
>
> **Why not simply use $I[f_\theta (x_i)>1-\tau]$?** While this can be used on the train data, this does not generalize well for test data. The underlying intuition is - While networks generalize predictions well, they do not generalize "confidences". Our approach essentially makes the network generalize both confidences as well as predictions.
>
> This can be observed in figure 1(b) as well. If we simply used $I[f_\theta (x_i)>1-\tau]$, we would get parallel lines (parallel to the base-classifier boundary) as the boundaries, but when you actually train you get "slanted" boundaries!
>
> (Another interpretation of algorithm 1) - One can think of $\tau$ as a proxy for the confidence value and by training (following algorithm 1) we are asking the network to learn - "Would you predict the label $1$ for sample $x$ with atleast probability $1-\tau$ or not?" Accordingly assign the probabilities (eq 10) for the test samples. These probabilities are more robust as illustrated with the experiments.
>
> **Figure (2)** The aim of figure (2) and the related experiment is to understand - What function does Algorithm 1 even learn? Our hypothesis is - It learns the direction in which the label changes (or confidence value changes)
>
> To verify this, we do the following experiment - Take a 1-d dataset in a 2-d space with two labels and learn the quantile representations $Q(x, \tau)$. Now design the labels as follows - At $\tau=\tau_0$, the labels would be $[1,1,\cdots,1 (\text{till } \tau_0), 0, 0, 0 \cdots]$. We ask the question - Which point $x$ in the original space would result in these labels at different $\tau$? This is learned using SGD on the input $x$ while fixing the network $Q(x,\tau)$ and the labels. The red dots are the one which we learn.
>
> While we do not get the original dataset, we do get a 1-d caricature of the original dataset. This shows that the quantile representations indeed learn the direction in which the label changes.
>
> **Comparision with deep-ensembles and DUQ** While deep-ensembles and DUQ also have the same effect of ability to identify training regions precisely, the approach is fundamentally different. DUQ computes the distance between centroids and the feature vectors in the representation space and accordingly assigns uncertainity. Deep ensembles do learn different classifiers, but each classifier has no discernable interpretation.
>
> Our approach on the other hand implicitly learns several boundaries in the input space, where each boundary has a specific meaning (confidence attributed to $\tau$). Accordingly, we assign confidence scores.

---

> > ### Comment · Reviewer_zXud · 2023-08-20
> > **Thanks for the clarification; however, my main concern still seems valid.**
> >
> > I read the other reviews and all rebuttals provided by the authors.
> >
> > I thank the authors for their answers to my questions. They actually clarified the points which were not clear to me from the original submission.
> >
> > However, my original concern, which is limited evaluation for out of distribution (OOD) detection still stands. Upon my review, the authors compared their OOD detection results with those of MLS, which was presented in ICLR2022. The new table in the rebuttal pdf shows that, in terms of OOD detection performance, the results are mixed. Sometimes the proposed QR method is better, sometimes the baseline is better. Actually, there are some interesting results: there are two cases where the proposed QR method outperforms the baseline with +16 and +14 percentage points, and there is one case where the baseline is better by 11 percentage points. These results needs careful review to explain what is going on. There might be some positive lessons to learn and consequently to improve the method.
> >
> > I believe it is the authors' responsibility to convince reviewers/readers about the merits of the new approach. The authors should have found an application area/problem where the proposed method is clearly superior to what is existing. There must be such a problem, right? Or something that previous methods could not even achieve but the proposed method does. Instead, OOD detection is presented as a major experiment, which results in "limited evaluation" opinions. The authors could still present OOD experiments as a sanity check, they should explicitly say that this is for sanity checking. And then, they could present an experiment where the proposed method clearly shows its advantage over what exists. The theoretical side of the paper is strong, so I believe there should be lots of experimental settings that shows the superiority of the proposed method.
> >
> > In addition, upon reading the other reviews and all rebuttals, it seems to be that there are a lot points that need clarification. So, the current presentation of the paper clearly needs a major revision.
> >
> > For reasons I explained above, in my opinion this paper is not yet ready for publication. Therefore, I am not changing my rating.

---

> > > ### Author Response · Authors · 2023-08-20
> > >
> > > We thank the reviewer for the response. And we completely agree that it is our responsibility to convince the readers the merits of the proposed approach.
> > >
> > > The main contribution of this article is qualitative. To our knowledge, there exists no method which can compute quantile representations of a pre-trained network without re-training from scratch. All methods we are aware of ignore the pre-trained network. We believe there to be a lot of practical value in this approach. Particularly the calibration experiment in the article shows that the quantile probabilities are distortion-invariant. Thus, if one wants to compute distortion invariant probabilities for a pre-trained network - This is not even possible with any of the existing approaches.
> > >
> > > And we also agree that the experiments were designed as sanity checks of the properties one would get if they computed quantile representations from scratch. So, we never claim that we are state-of-the-art, just that we are better than a strong MSP. baseline We also agree that the results are mixed with respect to MLS approach and hence our method is only comparable to it.
> > >
> > > Please note that, as stated in the MLS article referenced, to get the state-of-the-art results - scale, training strategies and hyper parameter tuning seems to be the key. However, reaching that scale requires a lot more novel ideas and compute power, which we could not get into due to computational limitations. Also to be fair, the ideas in the MLS article happened over a lot of years while this idea is novel and new.

---

### Author Rebuttal · Authors · 2023-08-04

**Evaluation:** We thank the reviewer zXud for pointing out MLS approach (ICLR2022). We include in the pdf the comparision between MLS and and maximum probability obtained using quantile representations. Unfortunately, MLS approach is not compatible with our approach since we have multiple boundaries and no unique way to compute the distance from the boundary or aggregate the distances from various boundaries. This is considered for future work. However, observe that even the maximum probability score for quantiles is better than the maximum logit score for resnet34/cifar10, and only drops a few points in resnet34/svhn configuration. Thus the approaches are still comparable.

Note that our aim was not to obtain state-of-the-art results, but rather provide sanity checking experiments validating our idea. There are several other aspects which can be included in our approach to achieve state-of-the-art : (i) Larger network for quantile representations than the base-network (ii) Better optimization by hyperparameter tuning (iii) Longer training etc.

Since the aim of this article is to validate the algorithm and present the duality idea, we have kept the base-network the same and only used simple optimization with little/no hyperparameter tuning.

**Figure 1 Explanation** The aim of the figure is to illustrate the intuition behind quantile regression for classification and motivate algorithm 1.

(1) We first identify a base-classifier $f_{\theta}(x)$ by simply using logistic regression between classes $1$ and $2$.

(2) Then (similar to algorithm 1) we construct a new dataset given a $\tau$ by thresholding the probability values. That is for a given $\tau_0$, construct the dataset $(\{x_i, I[f_{\theta}(x) > 1 - \tau_0]\})$ and train the logistic classifier on this data with $x_i$ as input and $I[f_{\theta}(x) > 1 - \tau_0]$ as expected output.

(3) Doing (2) for different $\tau_0$ results in different boundaries as you see in figure 1b. We have updated the figure 1b (see attached pdf) to include the values of $\tau$ for each classifier.

**From 1(b) to 1(c)** Let $x$ be any input, and say we get $y_{\tau}$ to be the corresponding logit for the classifier at $\tau$. Now, assign the representation - $[y_{0.1}, y_{0.2}, \cdots, y_{0.9}]$ to the input $x$. Use these representations (for each point in class 1 and 2) as input to OneClassSVM and identify the inliners vs outliers. The red region is marked shows inliners. This shows that the approach above can in-principle identify training region quite effectively.

**Wording of bullet 4 in algorithm 1:** Sorry for the confusion caused by the wording here. We hope to rephrase the line better as follows in the revised version:

*To get the classifier $Q(x,\tau)$, collect the dataset $D_{\tau}^{+} = \{(x_i, y_{i,\tau}^{+})}$ for all $\tau$, and train the classifier (DNN) using Binary Cross Entropy Loss with the new dataset. That is, for the input $(x_i,\tau)$ and the label predicted is supposed to be $y_{i, \tau}^{+}$.*

*NOTE:* We model $Q(.,.)$ using the same architecture as the pre-trained classifier but with additional input $\tau$.

**Alternate interpretation for Algorithm 1** While classical deep networks generalise predictions quite well, they are not capable to generalising confidences. Our approach solves this problem by asking the network to learn - " Would I predict the label $1$ for the input $x_i$ with probability at least $1-\tau$". This approach results in robust probabilities as illustrated with the experiments. Observe that here $\tau$ acts as a proxy for the confidence (which we obtain thanks to duality).

**References and Citations** We thank the reviewers for pointing out several different related works and shall include a more elaborate literature survey along with citations in the revised version.

---

### Decision · Program_Chairs · 2023-09-21

**Decision:**

Reject

**Comment:**

This paper tackles the problem of quantile regression (i.e., estimating several quantiles of the predictive distribution) for deep learning models. Using a duality, they manage to decouple the loss function from the pretrained model, which allows them to retrain the head only in order to estimate different quantiles. The approach is specific to binary classification.

While this is a potentially useful observation, reviewers feel that the empirical evaluation falls short of motivating the usefulness. Only a single baseline is compared against. Results for out of distribution (OOD) detection are mixed and not really conclusive. Moreover, the monotonicity of quantiles at each input is only encouraged, but not guaranteed. Also, more recent work on simultaneous quantile estimation in neural forecasting is not cited or compared against.